# Effects of Face Masks on Physiological Parameters and Voice Production during Cycling Activity

**DOI:** 10.3390/ijerph19116491

**Published:** 2022-05-26

**Authors:** Arianna Astolfi, Giuseppina Emma Puglisi, Louena Shtrepi, Paolo Tronville, Jesús Alejandro Marval Diaz, Alessio Carullo, Alberto Vallan, Alessio Atzori, Ada Ferri, Francesca Dotti

**Affiliations:** 1Department of Energy, Politecnico di Torino, Corso Duca degli Abruzzi 24, 10129 Torino, Italy; giuseppina.puglisi@polito.it (G.E.P.); louena.shtrepi@polito.it (L.S.); paolo.tronville@polito.it (P.T.); jesus.marval@polito.it (J.A.M.D.); 2Department of Electronics and Telecommunications, Politecnico di Torino, Corso Duca degli Abruzzi 24, 10129 Torino, Italy; alessio.carullo@polito.it (A.C.); alberto.vallan@polito.it (A.V.); alessio.atzori@polito.it (A.A.); 3Department of Applied Science and Technology, Politecnico di Torino, Corso Duca degli Abruzzi 24, 10129 Torino, Italy; ada.ferri@polito.it (A.F.); francesca.dotti@polito.it (F.D.)

**Keywords:** breathing resistance, face masks, filtration efficiency, sports performance, voice analysis

## Abstract

This study investigates the effects of face masks on physiological and voice parameters, focusing on cyclists that perform incremental sports activity. Three healthy male subjects were monitored in a climatic chamber wearing three types of masks with different acoustic properties, breathing resistance, and air filtration performance. Masks A and B were surgical masks made of hydrophobic fabric and three layers of non-woven fabric of 100% polypropylene, respectively. Mask S was a multilayer cloth mask designed for sports activity. Mask B and Mask S behave similarly and show lower sound attenuation and sound transmission loss and lower breathing resistance than Mask A, although Mask A exhibits slightly higher filtration efficiency. Similar cheek temperatures were observed for Masks A and B, while a significantly higher temperature was measured with Mask S at incremental physical activity. No differences were found between the masks and the no-mask condition for voice monitoring. Overall, Mask B and Mask S are suitable for sports activities without adverse effects on voice production while ensuring good breathing resistance and filtration efficiency. These outcomes support choosing appropriate masks for sports activities, showing the best trade-off between breathing resistance and filtration efficiency, sound attenuation, and sound transmission loss.

## 1. Introduction

The spreading of the COVID-19 virus has changed habits and lifestyles. Most industrialized countries have made it mandatory to wear face masks indoors and outdoors to reduce infections; however, the progress of the pandemic has evolved over time, and such requirement turned out to be not compulsory in some settings—especially those involving open-air individual sports activities. It has been noticed in many circumstances at the first stages of the pandemic that the use of face masks significantly influences speech communication. Among other scenarios, face masks affect voice production also during sports activities at different intensity levels, especially those in which the use of such individual protection is suggested, such as team sports that imply the interaction between athletes. This issue is unexplored, although it deserves attention as a condition with a risk of infection.

In sports activities, it is suggested to use personal protective equipment. In order to be effective, it should be characterized by enhanced breathing resistance and filtration performances and characteristics that ensure good speech intelligibility. During sporting activities, indeed, instructors and athletes usually raise their voices to improve speech comprehension [1], and this implies the incurrence of occupational hazards [2,3,4] that can be even more significant when face masks are worn [5]. Furthermore, speech intelligibility decreases more in the case of dysphonic voice [6].

A new face mask for sports activities was designed and tested at Politecnico di Torino, and its performance was compared to those of commercial surgical masks. As speech communication is essential in team sports, e.g., basketball, but field tests can be affected by the environment, this study was carried out under controlled ambient conditions. To this aim, the cycling activity was monitored in the laboratory to improve the repeatability and reproducibility of the experiments. From a communicative point of view, cyclists need to warn the whole team to plan the upcoming steps in the run. At the same time, as ventilation increases during the sports activity [7], reaching values up to (100–200) L/min compared to 6 L/min at rest, optimal breathing conditions are needed. Thus, a good mask for sports practice should enhance breathing resistance performance while maintaining adequate filtration efficiency.

As far as the acoustic scenario is concerned, face masks affect speech communication, leading to excessive vocal effort for the talker and increased listening effort for the listener [5,8] due to several factors. Among others, it is worth considering: (i) the use of face masks, which differ in material and shape, (ii) the communication intent and situation (e.g., talking with a spontaneous or a clear speech [9]), (iii) the characteristics of talkers and listeners (e.g., having or not having voice pathologies [6], using or not using a voice amplification system [10], being or not being either normal hearing or mother-tongue [11]), (iv) the acoustic environmental conditions (e.g., noise and reverberation [12]). However, for the aim of the present pilot study, the latter was not considered.

Further than the effects on vocal production are concerned, the use of face masks also affects physiological parameters, as shown in a recent review by Litwinowicz et al. [13]. Even though some works have shown that heat stress is a significant concern when wearing a face mask, regardless of the performed activity, poor information is available about the effect of a face mask during amatorial or professional sports practice. Most studies involved healthcare professionals in hospital settings [14]. Epstein et al. [15] showed a mild but significant increase in CO_2_ partial pressure when an intense cycling activity is performed wearing an N95 respirator face mask, which has reduced breathability but higher filtration efficiency with respect to surgical masks. Ruth et al. [16] demonstrated that the microclimate in a face mask affects the overall thermal sensation. Li et al. [17] compared N95 and surgical masks in heart rate, heat stress, and subjective sensations during an intermittent exercise on a treadmill. It was observed that N95 face masks have a higher impact than surgical masks on heart rate and thermal stress, but no comparison with and without a mask was carried out. Person et al. [18] investigated the effect of a surgical mask on a 6 min walking activity. They reported that the mask did not alter heart rate, saturation, and distance, while dyspnea variations were clinically significant. Physiological variations when using face masks whilst performing low-intensity physical activity were found to be reduced when wearing a surgical mask with respect to an N95 mask, as shown by Litwinowicz et al. [13].

The materials and shapes of face masks impact their acoustic and air filtration performances. Previous studies adopted different methodologies to establish the acoustic properties of face masks; thus, a large variability across results may be found [19]. Typically, opaque masks are made of materials with a porous structure of natural or synthetic fibers. Hence, they mainly absorb sound energy in the high-frequency range between 2 kHz and 8 kHz [8,19]. The absorption turns into a strongly detrimental effect on speech intelligibility, which mainly depends on mid-to-high frequencies from 0.5 kHz to 4 kHz [20]. From an air filtration point of view, the number of layers and the type of fibrous material affect the mask performance. The initial filtration performance is expressed by airflow resistance and filtration efficiency [21] of the clean face mask. Airflow resistance of face masks is typically expressed as breathing resistance [22]. The filtration efficiency is a function of the size of the particles to be captured [21]. The electrostatic charges on fibers and particles strongly influence the efficiency when particle size is below 1 μm [21]. Liquid and solid particles of the same size show different removal efficiencies when the velocity of the air through the filter material is above 20 cm/s [23]. The number distribution of particles emitted by humans as a function of their size when people cough, sneeze, or speak has a peak of around 1 μm [24,25]. The peak of the volume distribution as a function of the particle size shifts towards larger sizes since volume and mass depend on the cube of the particle size. Hybrid fabrics and densely woven cotton are characterized by a higher filtration efficiency than loosely woven cotton masks [26]. Three-layer masks were found to have a filtration efficiency similar to synthetic surgical masks [27].

This study investigates several domains that affect voice production, focusing on cyclists that perform incremental sports activity while wearing masks. The need to guarantee good speech communication while performing team sports was a question of the presented exploratory study. Then, together with vocal parameters, physiological ones were also investigated as they were found to be significantly influenced by the use of face masks. In order to corroborate these issues, three face masks were studied: two general-purpose surgical masks and one specifically designed for sports. The reason for selecting these typologies of face masks is that the two surgical ones are easily accessible in everyday life and are typically used also to perform sports activities [28]. Indeed, during the pandemic, the Italian authorities indicated these devices as adequate to reduce the spreading of the virus. As they are readily available but less effective in removing particles than N95 respirators, it was decided to evaluate the effects of these mask typologies on voice production and physiological outcomes. As far as the sports mask is concerned, it was designed ad hoc within a research and innovation collaborative project among several Departments of Politecnico di Torino. All of them were characterized by acoustic properties, breathing resistance, and filtration efficiency. Three cyclists were monitored with a contact-sensor-based microphone placed at the base of the neck, a wireless temperature and humidity sensor placed on the cheek, and a heart rate sensor placed on the chest. The cyclists carried out incremental physical exercises in a climatic chamber while performing a speech task aloud. The tests measured the average temperature and humidity of the skin on the cheek, the heart rate, the rate of perceived exertion, and the vocal parameters. The outcomes of this work constitute a baseline to draw an appropriate trade-off between mask performances in terms of breathing resistance and filtration efficiency and an acceptable degree of sound absorption to identify masks that are suitable for sports activities.

## 2. Materials and Methods

### 2.1. Face Masks Selection

Three face masks were considered that are shown in Figure 1: two certified EN 14683 [29] standard surgical masks (A and B), each composed of three layers, and a cloth mask with a disposable filter (S), which was specifically designed for sports. The surgical masks A and B were selected as they (i) are easily accessible in everyday life; (ii) resulted in having an adequate filtration efficiency in agreement with EN 14683 [29], as detailed in the next paragraphs; (iii) significantly differed in breathing resistance performances; and (iv) significantly differed in acoustic attenuation performances.

Mask A is made of hydrophobic fabric. Mask B is composed of three layers of non-woven fabric of 100% polypropylene, and Mask S is composed of a reusable frame made of polyester mesh fabric Panatex, AT1410, and elastic Lycra strips to keep the mask in place. A disposable filtering non-woven fabric is inserted in this knitted elastic fabric pocket. The disposable filter placed in Mask S comprises two hydrophilic spun-bond PP layers and a middle nanofiber layer (UFI filter, [30]). The average weights are (5.20 ± 0.02) g for Mask A, (3.20 ± 0.02) g for Mask B, and (21.3 ± 1.4) g for Mask S.

### 2.2. Participants, Test Facility, and Organization

Three healthy male subjects, whose biometric data are summarized in Table 1, performed physical activity tests while describing a map to an experimenter about 3 m in front of them (Figure 2), and they were paid for their participation. The testing procedures agreed with the ethical standards of the Declaration of Helsinki. The subjects were non-professional cyclists. Their functional threshold power (FTP), i.e., the maximum power the subject was able to sustain over one hour period on the bike [31], was estimated during a preliminary test on a smart trainer (Drivo 2, Elite, Italy), according to the procedure implemented in the software My E-trainer (Elite, Italy).

The physical intensity of the smart trainer was increased stepwise. Phases 1, 2, and 3 lasted from minutes 0 to 10, 10 to 20, and 20 to 30, respectively. They were characterized by intensities of 40% FTP, 80% FTP, and 120% FTP, respectively. Each test thus lasted about 30 min overall. Each subject carried out four tests, one with each face mask and a reference test without a face mask. Overall, twelve tests were completed. The subjects wore the same outfit in all the tests, consisting of cycling shorts and a short-sleeve shirt, and performed one test per day at the same hour to avoid bias due to circadian rhythm.

The test order was randomized according to a complete design of experiments. The heart rate was monitored continuously with a Garmin heart rate monitor placed on the chest. To monitor skin temperature and humidity, which are considered in the literature as relevant parameters to assess thermal stress [16], a wireless temperature and humidity sensor (I-button DS1923) was placed on the cheek on the area covered by the face mask.

Tests were carried out in a climatic chamber (Befreezer, Italy) of 56 m^3^ volume with air temperature Ta = (22.2 ± 0.6) °C and relative humidity RHa = (57.5 ± 3.6)%. The following physical parameters were acquired: (i) the average skin temperature on the cheek, (ii) the skin relative humidity on the cheek, (iii) the heart rate, and (iv) the mask wettability. During the tests, subjective sensations about thermal and ergonomic comfort were collected through a questionnaire. The bipolar scale of thermal assessment (cold, cool, slightly cool, neutral, slightly warm, warm, hot) was given by the international standard EN ISO 28802:2012 [32]. Furthermore, the Borg scale [33] was used to quantify the perceived exertion during physical activity on a scale from 6 (no exertion at all) to 20 (maximum exertion). Subjective surveys were collected at the end of each phase, resulting three times during the test at three different intensity levels. Face masks were weighed before and after the test on a balance (uncertainty of 1 mg) to quantify the amount of sweat and perspiration residue. Free comments about perception and judgment about the comfort of the face mask were allowed and recorded at the end of the test.

### 2.3. Measurement of Vocal Parameters

#### Voice Monitoring Device

In order to avoid the effect of the environment on voice signal acquisitions and to allow for wearability of a monitoring device within the activity performance, as suggested in the literature [34,35], a portable contact-sensor-based system was used to record voice production in this study. Indeed, such an approach guarantees the stability of the microphone and excludes any effect in the vocal analysis due to the environmental noise. The accuracy of voice measures performed with contact-sensors-based devices was investigated in the literature [36]. As far as the equivalency of the voice parameters that are obtained with the two devices, Astolfi et al. [37] reported that for the mean and equivalent SPLs, an air-microphone provides a lower uncertainty (~2 dB) than the contact-sensor-based device (~3 dB). However, using the latter is advantageous despite its higher uncertainty, as the air-microphone could be affected by other acoustic sources, physical activity, distance, and orientation from the mouth during the tests. Within this experimental campaign, voice monitoring was thus performed using the Vocal Holter Med (VHM) device, based on a piezoelectric contact microphone (model HX-505-1-1, HKKK, Jiadind Science Park, Dalang, Longhua New Dist., Shenzhen, Guangdong, China) positioned at the neck with a collar and connected to a data logger. The contact microphone is insensitive to the background noise in the room where the monitoring is carried out [35,38,39]. Moreover, the previous studies [35] investigated the reliability of voice monitoring results based on workers performing their everyday activities, including body movements. The microphone has a not negligible sensitivity to body movements, but their effects are made negligible by means of a digital low-pass filter (implemented in the VHM firmware) that is able to separate the frequency components related to the voice activity from the ones related to the body movements. The VHM device acquires signals as the vibration of the vocal folds while phonating at a sampling frequency of 22,050 Sa/s. The measured samples are then grouped in frames of 46.4 ms (1024 samples), as this value is of the same order of magnitude as the average inter-syllabic pause for the Italian language.

The VHM estimates voiced sound pressure levels (SPLs, in dB) at a fixed distance from the mouth after calibration was carried out. It particularly relates the voltage levels measured at the output of the contact microphone to the SPLs acquired by a calibrated air microphone 22 cm far from the mouth, on axis [39]. In order to normalize the SPLs, the value estimated at 22 cm was converted into vocal effort levels at 1 m from the mouth, applying the relation for sound propagation in the free field. Based on the previous literature related to voice monitoring with VHM [40,41,42,43], beyond the sound pressure level (SPL_1 m_, dB), the fundamental frequency (F_0_, Hz) and the phonation time percentage (D_t_, %) were extrapolated from the speech task. From the distribution of occurrences of SPL1 m and F0, the main statistical quantities were obtained, which are mean (F_0mean_, SPL_mean_), median (F_0med_, SPL_med_), mode (F_0mode_, SPL_mode_), and standard deviation (F_0stdev_, SPL_stdev_).

### 2.4. Voice Monitoring Procedure

Voice monitoring consisted of a long-term evaluation of voice while cycling at different FTP and wearing or not a face mask. Such long-term evaluation of voice was assessed by describing a map, i.e., a form of natural speech that evokes a specific communication intent [9]. Each tester described four maps, one for each of the three masks and one for the no mask condition, to an experimenter standing about 3 m in front of him with communicative intent. Each map included 12 “landmarks” (e.g., rocks, lake, school bus), a path connecting the starting and ending points, and it was considered equivalent to the others in terms of items and degree of difficulty [44].

Voice monitoring was performed with the VHM, as shown in Figure 3. In the climatic chamber, noise from ventilation equipment was present randomly during the measurements. It was a constant low-frequency noise, but it did not affect the measurements with the VHM as previously indicated.

### 2.5. Acoustic Characterization of the Face Masks

Face masks were tested concerning their sound attenuation when worn by a B&K Head and Torso Simulator (HATS) in the anechoic chamber of Politecnico di Torino, as shown in Figure 4 for Mask S.

Sound attenuation was evaluated as the difference in the Sound Pressure Level (SPL) at 1 m from the HATS mouth with and without the mask (∆SPL). Measurements were carried out using a white noise signal, and results were analyzed in one-third of octave bands as averages between those bands that are more important for speech intelligibility [20,45,46], from around 2 kHz to 4 kHz. We included anyway two different frequency ranges from 0.4 kHz to 5 kHz and from 1.6 kHz to 5.0 kHz, respectively, to have a clear picture of the performance of the different masks in broader ranges. The normal-incidence absorption coefficient (α_0_) and the sound transmission loss (STL) were also measured for the three filtering materials in the impedance tube according to ISO 10534-2 [47] and ASTM E1050-19 [48] for the former, and ASTM 2611 [49] for the latter. The reason why the absorption coefficient was measured stands in the objective of considering the property of a mask’s material as a design parameter that can be easily evaluated to predict the final acoustic performance. This can be assumed in a similar way as the breathing resistance and filtration efficiency, so an additional parameter can be used to predict the degree of protection performance.

Measurements were based on the impedance tube HW-ACT-TUBE and -STL (Siemens, Munich, Germany), which has a diameter of 35 mm and is equipped with two ¼’’ flush-mounted GRAS 46BD (GRAS, Holte, Denmark). The method allows accurate sound pressure amplitude and phase measurements in the broad frequency range of interest, i.e., (0.1–5) kHz [47]. A white noise source (2” aluminum driver) integrated with the system produces stationary high sound levels (100 dB) inside the tube, ensuring a high signal-to-noise ratio and a flat spectrum in the frequency range of interest. The sample holder consists of a tube with a rigid sliding piston. The method allowed a higher control on the positioning of the different samples with limited thickness into the flanged-to-sample holder of the tube. In this way, the size of the voids between the tested material and the sample holder was reduced. Hence, the circumferential effect discussed in [48] could be neglected. The impact of the irregularities in the samples was considered by repeating the tests for each material with three different samples, evaluating the reproducibility uncertainty contribution. Temperature and atmospheric pressure were measured and considered in the absorption coefficient measurements.

The sound transmission loss (STL) was measured using an extension of the previously described impedance tube (HW ACT-STL) [50], which was equipped with three microphone holders placed at a 65 mm and a 29 mm spacing. As stated in ASTM 2611-09, for a single transfer matrix measurement, or sound transmission loss measurement, two basic measurements with two different terminations were involved, i.e., anechoic and open-ended tube. Finally, the measured results were processed as fine frequency average values (2 Hz resolution) and standard deviations over the highest frequencies that are more important for speech intelligibility, i.e., from about 1 kHz to 5 kHz [20,45].

### 2.6. Measurement of Air Filtration Performance

An accurate characterization of the air filtration performance of face masks is of utmost importance to obtain useful data. Norms standardize optimal thresholds, particularly as far as the breathing resistance and filtration efficiency are concerned; therefore, these two parameters were considered to qualify the face masks.

Filtration performance can be assessed by testing a flat piece of filter material or a full-scale sample. The choice between the two approaches depends on whether a part of the filter material can characterize the performance of the full-scale mask. For the aim of the present study, only flat samples were tested, and the air filtration performance was measured according to the procedures described in the following sections.

#### 2.6.1. Breathing Resistance

Breathing resistance is a measure of the resistance to airflow. Different approaches can be used to measure it, depending on the type of face mask being assessed, and there is a lack of information regarding the measurement uncertainty on the standards [51]. In the case of flat samples, it can be assumed that the airflow is distributed across the whole test surface, and valid results are ensured when the same layers make up the entire mask across the surface. Hence, airflow resistance is proportional to the air velocity passing through the filtering area of the flat sample (obeying Darcy law [52]). By using a Textest FX 3300 LabAir IV, these measurements were carried out by setting the face air velocity crossing the filter media. As prescribed by UNI/PdR 90-2:2020 [53], the resistance to airflow was measured at 27.2 cm/s using a head adapter with a surface of 5 cm^2^.

#### 2.6.2. Filtration Efficiency

Filtration efficiency was measured as a function of the particle size for each filtering material, obtaining a fractional efficiency curve. The airflow was processed through the test rig to meet the data quality requirements specified in UNI/PdR 90-2:2020 [53]. A HEPA filter completely removed the particles present in the intake air to guarantee that the particles generated by the synthetic aerosol generator were the only ones measured for the efficiency assessment.

The synthetic aerosol was made of liquid spherical particles of DiEthylHexylSebacate (DEHS) produced by a Laskin nozzle. The test aerosol was injected into a plenum, then mixed into near-uniform concentration by passage through an upstream and a downstream orifices plate of the test sample section. The test aerosol was primarily intended for efficiency measurements in size range from 0.3 µm to 3.0 µm, extending up to 10.0 µm. The test section maintained overpressure (from 200 Pa to 500 Pa) using two fans. By controlling the internal pressure of the test duct, it was possible to avoid ambient particles leaking into it.

The filtration efficiency (*E_ps_*) of the test sample is equal to 1 minus the penetration (*P_ps_*), i.e., the number of particles passing through the test device section, and then reported as a percentage. The *P_ps_* can be obtained as the downstream and upstream concentration ratio in each particle size interval.

The filtration efficiency by particle size was calculated by knowing the particle concentration of the synthetic aerosol, which was sampled upstream and downstream several times during each test. In order to measure the number concentration as a function of the particle size, we used a single optical particle spectrometer (TSI 3330) based on the light scattering principle. An auxiliary pump was used to guarantee an isokinetic sampling within 10% of the air velocity in the duct.

The upstream and downstream sampling lines had identical geometry to guarantee equal particle losses by aerosol transportation. The impact of different particle losses was corrected through the correlation ratio test. The number of particles was measured as a function of their size upstream and downstream of the test device section with the aerosol generator turned on and without any device in the test section. The data obtained with the correlation ratio test corrected the filtration efficiency data following the mathematical procedure given in Section 9.3.3.2 of UNI/PdR 90-2:2020 [53], as the optical particle spectrometer provided the number of particles for several “channels” covering the whole measured size spectrum from 0.3 µm to 10.0 µm.

In order to characterize the filtering performance of face masks with a single number, the UNI/PdR-90 defines the term *eCFC*, which is related to the face mask’s particle removal efficiency. For this aim, the following assumptions were made: (i) the particle size range of interest was (1–3) μm, (ii) the *eCFC* was calculated considering the upstream number of particles equal in each channel; (iii) the width of each size channel was used to weight the efficiency. The wider the size channel, the higher contribution to *eCFC*. Fractional efficiency *eCFC* was obtained from Equation (1):(1)$$eCFC=\sum_{i=1}^{n}E_{i}\;*q_{3}\left(\bar{d_{i}}\right)*\Delta \ln\left(\bar{d_{i}}\right)/\;\sum_{i=1}^{n}q_{3}\left(\bar{d_{i}}\right)*\Delta \ln\left(\bar{d_{i}}\right)$$
where *i* is the index of the size channel which covers the particle size range between 1.0 μm and 3.0 μm, *n* is the total number of size channels that are used in the calculation of *eCFC*, *E_i_* is the filtration efficiency for the size channel i, *q_3_* is the particle volume as a function of their size, $\left(\bar{d_{i}}\right)$ is the geometric mean, and Δln(*d_i_*) is defined as in Equation (2):(2)$$\Delta \ln\left(\bar{d_{i}}\right)=\ln\left(d_{i+1}\right)-\ln\left(d_{i}\right)=\ln\left(d_{i+1}/d_{i}\right)$$

### 2.7. Statistical Analysis for Differences among the Masks

Statistical analyses were carried out with the IBM SPSS statistics package (version 21.0, Armonk, NY, USA) to investigate differences among the masks. The Kruskal–Wallis statistical test (KW) for non-parametric data distributions was applied to search if differences among the masks were significant for each parameter [54]. It is used when more than two distributions are compared, but it does not identify the different samples. The Mann–Whitney U test (MWU) was then applied to test the pairs of masks to determine which differed from the others.

## 3. Results and Discussion

### 3.1. Face Masks Characterization

Table 2 shows the acoustic and filtration performances of the face masks in terms of sound attenuation ∆SPL in the third-octave bands’ ranges of (0.4–5) kHz and (1.6–5) kHz, sound absorption (α_0_) in the third-octave bands’ ranges of (1–5) kHz and (0.9–3) kHz, sound transmission loss (STL) in the third-octave bands’ ranges of (1–5) kHz and (1–3) kHz, breathing resistance and filtration efficiency (*eCFC*). ∆SPLs were evaluated with the Head and Torso Simulator in the anechoic room; thus, no averaging was made considering the tester. For the sake of completeness, ∆SPLs are also given as third-octave bands spectra in Figure 5.

Table 2 also shows the lowest (better) ∆SPL and STL values for Mask S compared to the other masks, while Mask A has the highest values. More similar is α_0_ across the three masks, with the lowest value for Mask B. Mask A shows the highest (better) *eCFC* values, while Mask S provides the lowest value. The same result is valid for fractional efficiency, higher for Mask A and lower for Mask S, as shown in Figure 6.

### 3.2. Voice Monitoring

The voice sound pressure level (SPL) was measured during the long-term monitoring. The environmental noise present during the tests did not affect SPL thanks to the high insensitivity of the contact microphone with respect to the pressure in the air [35]. Figure 7 shows the mean SPL related to each long-term test averaged over the three subjects for each physical effort and the four tested conditions (no mask and Masks A, B, and S). Vertical bars represent the expanded uncertainty (95% of confidence level, C.I.), which considers the instrumental uncertainty of the VHM, the contribution related to the fitting of the calibration curve [38,39], and the subject reproducibility.

An increasing trend appears as the physical effort increases, but the measurement uncertainties are overlapped, and no significant differences can be assessed. It can be assumed that the presence of the face masks does not imply, itself, an increase in vocal effort. However, this outcome still needs to be deepened due to the small number of the involved testers and would also require the monitoring of a rest and comfortable speaking condition to be compared.

The phonation time percentage (Dt%) was also evaluated. The results are shown in Figure 8. Albeit there is no significant difference between the Dt% values considered at different physical stresses, a descending behavior can be noted as the physical effort rises. This effect is due to the reduction in vocalization duration concerning the silence between syllables and words. The lengthening of silence duration is due to the necessity of breathing between words to keep up with the requested physical task.

Other statistical measures related to the SPLs distribution and the fundamental frequency did not exhibit statistical relevance to the study.

### 3.3. Sport Activity Tests and Physiological Outcomes

#### 3.3.1. Cheek Skin Temperature and Humidity

Average cheek temperatures and relative humidity with standard deviations (shadowed areas) are reported in Figure 9.

Skin temperature under the mask increased by (3–4) °C compared to bare skin. Similar temperatures were observed for Masks A and B, while a higher temperature was measured with Mask S, along with the three phases of incremental physical activity, i.e., (0–10) min, (11–20) min, (21–30) min, respectively. This effect can be due to the cloth mask shape that seals the face through elastic straps, while surgical masks allow for lateral air circulation. Mask S was designed for sport, and a firm attachment to the face was needed to avoid mask misplacement during contact between athletes. Relative humidity on the cheek was higher when wearing a mask than on bare skin. However, no significant differences were observed across masks, and values were always close to saturation during intense phases when sweating was abundant. These results are consistent with the literature, which shows the higher temperature for a sealing mask-like N95 compared to surgical masks [16].

#### 3.3.2. Mask Wettability

The weight of the masks at the end of the test, displayed in Table 3, shows that the surgical masks (A and B) retain less moisture than the cloth mask (S), even though the relative weight increase is more relevant for the lightest mask.

#### 3.3.3. Heart Rate

The heart rate during the three phases of the test is shown in Figure 10. Although the heart rate increased progressively with the physical effort, the average heart rate was not affected by wearing a mask. No significant differences were observed between the reference test without a mask and any tests carried out with a mask. It was about 100 bpm for Phase 1, about 120 bpm for Phase 2, and about 150 bpm for Phase 3. Even though masks are suspected of impairing gas exchanges and potentially causing hypercapnia, the athletic performance was not affected by the mask in this specific submaximal test. Although the amount of data in this field is quite limited, this result is consistent with those reported by Person et al. [18], who did not observe variations in heart rate when wearing a face mask during a walking exercise.

#### 3.3.4. Subjective Sensations

Figure 11 shows the thermal sensation before starting and at the end of each phase. Albeit few subjects were involved, the perceived thermal sensation was affected by physical activity intensity but not correlated with the mask use, as uncomfortable hot thermal sensations were expressed during the most intense phases either with or without a mask.

Humidity sensations are summarized in Figure 12. Uncomfortably humid sensations were perceived during intense phases with and without a mask, suggesting that all masks can be used without affecting thermal comfort. The perceived exertion shown in Figure 13 confirmed that the masks played a negligible role in the fatigue experienced. However, when asked “*Do you feel constriction due to the mask?*” and “*Do you feel fatigued because of the mask?*”, subjects expressed dissatisfaction with the mask use. Nevertheless, this psychological condition did not affect their perceived physical performance.

### 3.4. Overall Statistical Evaluation of the Three Masks

The effect of different face masks was investigated in an overall analysis considering all the measured parameters (i.e., for mask characterization, physiological, vocal) as influenced quantities. According to the KW non-parametric test, the parameters ∆SPL_0.4–5 kHz 3rd oct_, ∆SPL_1.6–5 kHz 3rd oct_, α0_1–5 kHz_, α0_0.9–3 kHz_, STL_1–5 kHz_, STL_1–3 kHz_, breathing resistance and *eCFC*, resulted in being different among the face masks (*p*-value < 0.05). The MWU test revealed that all the face masks were statistically different (Z = −2.236, *p*-value = 0.025). The presence or absence of a mask influenced physiological measures. The KW test revealed a difference for the cheek temperature in phase 2 (*p*-value = 0.041) and in phase 3 (*p*-value = 0.025). The MWU test revealed statistically significant differences concerning the cheek temperature in phase 2 and phase 3 under Mask A and Mask S (Z = −1.964, *p*-value = 0.05) and under Mask B and Mask S (Z = −1.964, *p*-value = 0.05). Masks B and S show statistically different cheek temperatures for the no mask condition in phase 2 and phase 3, whereas mask A exhibits such difference only in phase 3 (Z = −1.964, *p*-value = 0.05). Vocal parameters were not significantly influenced by the face mask (*p*-value > 0.05).

## 4. Conclusions

This study investigated the external domains that affect voice production, focusing on cyclists that performed three phases of incremental physical activities with different face masks. Three healthy non-professional male cyclists were monitored in a climatic chamber without a mask and wearing two typical surgical masks (A and B) and a newly designed cloth mask (S). Mask A is made of hydrophobic fabric. Mask B is composed of three layers of non-woven fabric of 100% polypropylene. Mask S is a multilayer cloth mask specifically designed for sport. Cyclists were asked to describe a map to an experimenter standing 3 m in front of them while quantities related to voice production, physiological parameters, and perceived exertion were acquired. The main conclusions are highlighted hereby:

Masks B and S behave similarly and better (lower) in terms of sound attenuation, sound transmission loss, and breathing resistance than Mask A, although mask A performs better (higher) for filtration efficiency, but not to a larger extent;Skin temperature under the mask increased by (3–4) °C compared to bare skin. Similar temperatures were observed for Masks A and B, while a significantly higher temperature was measured with Mask S in the last phases of incremental physical activity. However, this does not reflect in the correspondent’s subjective perception;No differences were found in the voice monitoring due to the face masks compared to the no mask condition.

Because of the latter conclusion, which reveals a homogeneity of response in terms of voice production when wearing one of the three face masks, it can be concluded that Masks B and S are suitable for performing team sports because (i) they do not show adverse effects on speech communication, (ii) ensure better breathing resistance, and (iii) guarantee good filtration efficiency. Although Mask A presents a higher filtration efficiency, indeed, it has lower breathing resistance and may thus be less suitable for the performance of sports activities that imply an increased physical exertion. One should note that the results refer to a small sample of subjects due to the complex experimental protocol. Authors will plan experiments to overcome this limitation and validate the outcomes, also considering other testers and test facilities.

## Figures and Tables

**Figure 1 ijerph-19-06491-f001:**
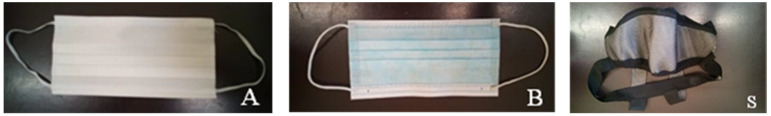
Face masks tested in the study: Masks (**A**,**B**) are two surgical masks, Mask (**S**) is an ad hoc designed mask for sport.

**Figure 2 ijerph-19-06491-f002:**
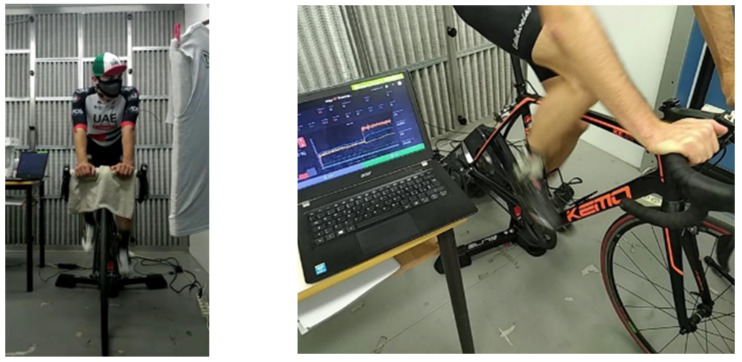
Experimental setup.

**Figure 3 ijerph-19-06491-f003:**
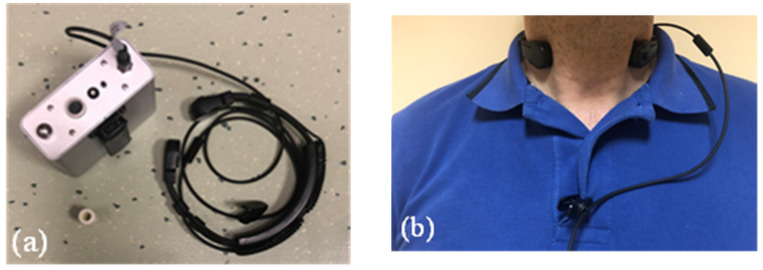
Vocal Holter Med (VHM) device (**a**) with a focus on the positioning of the contact microphone positioned at the neck by means of a collar (**b**).

**Figure 4 ijerph-19-06491-f004:**
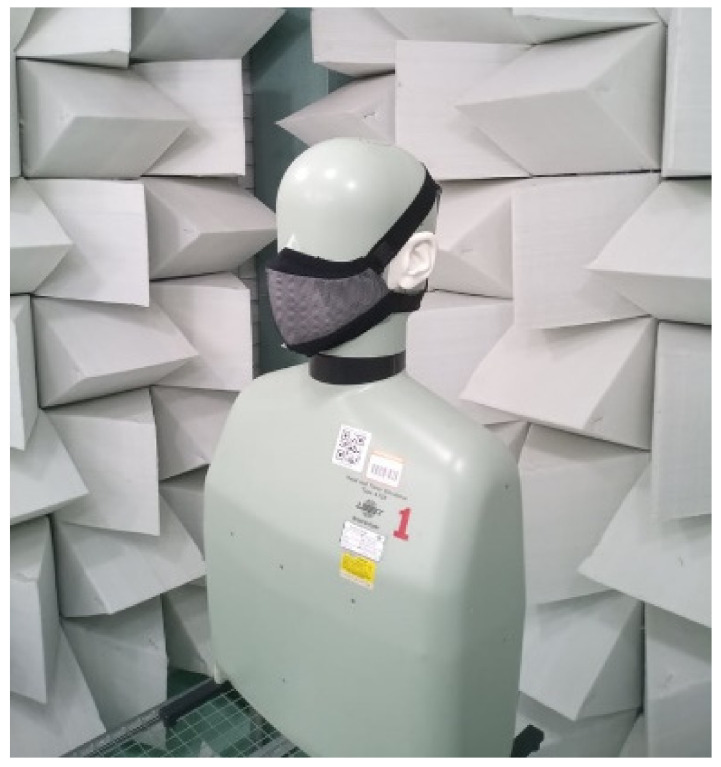
Mask S on the Head and Torso Simulator in the anechoic chamber of Politecnico di Torino.

**Figure 5 ijerph-19-06491-f005:**
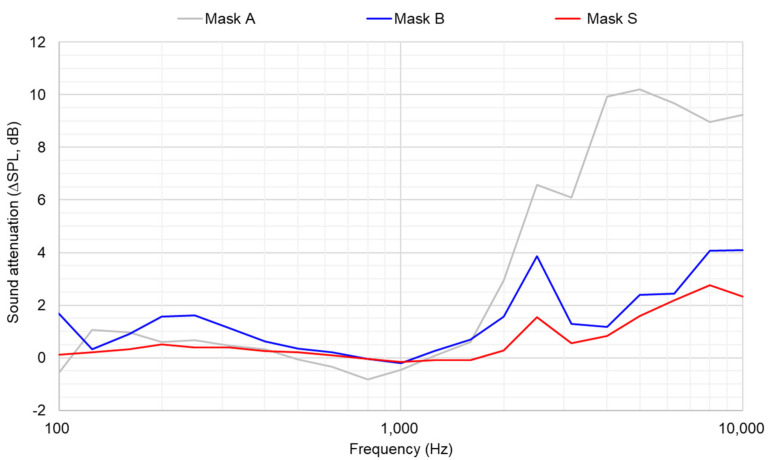
Spectral distribution of sound attenuation (∆SPL) for the three face masks.

**Figure 6 ijerph-19-06491-f006:**
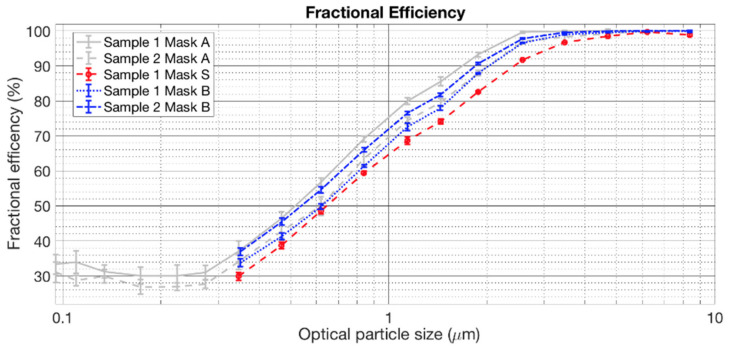
Fractional efficiency for each sample tested and for each mask.

**Figure 7 ijerph-19-06491-f007:**
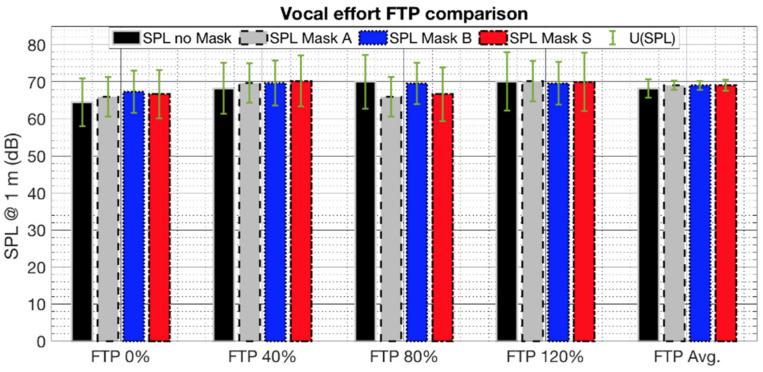
Averaged sound pressure levels at 1 m from the mouth (SPL@1 m) for different mask conditions, reported as averages over the three subjects for the four efforts (FTP at 0%, 40%, 80%, 120%) and as overall (FTP Avg).

**Figure 8 ijerph-19-06491-f008:**
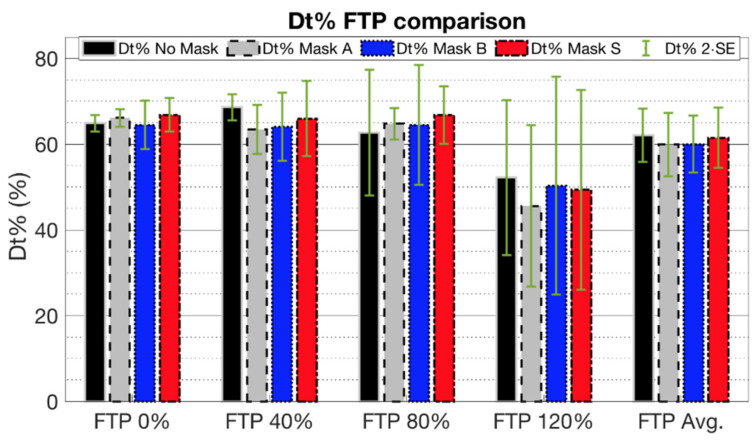
Averaged phonation time percentage (Dt%) for the different mask conditions, reported as averaged over the three subjects for the four physical efforts (FTP at 0%, 40%, 80%, 120%) and as overall average (FTP Avg).

**Figure 9 ijerph-19-06491-f009:**
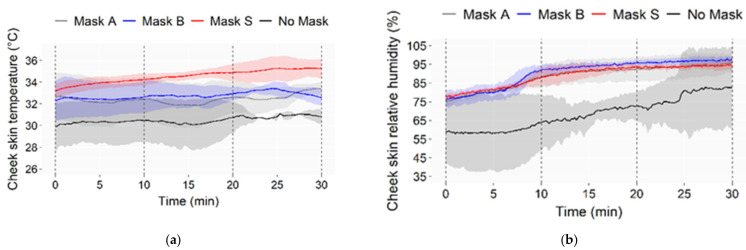
Cheek skin temperature (**a**) and relative humidity (**b**) reported as averages across testers. Shadowed areas represent the standard deviations.

**Figure 10 ijerph-19-06491-f010:**
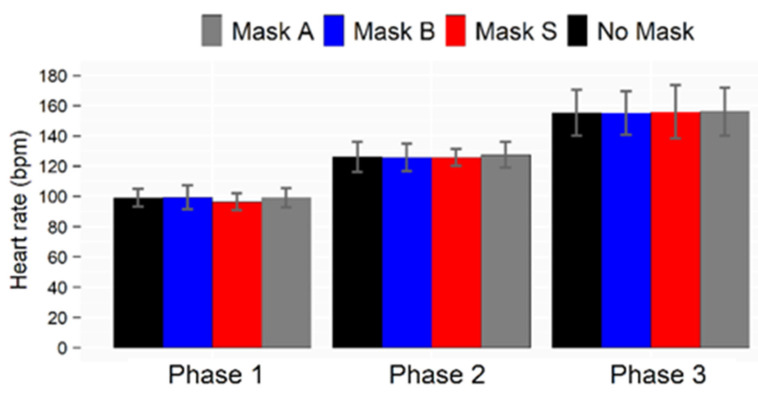
Heart rates measured during the physical activity phases and reported as averages across testers. Error bars refer to the standard deviations.

**Figure 11 ijerph-19-06491-f011:**
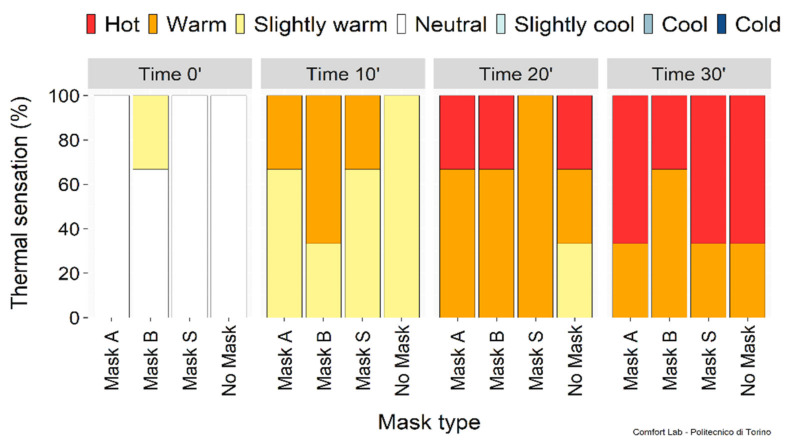
Perceived thermal sensation during the physical activity phases.

**Figure 12 ijerph-19-06491-f012:**
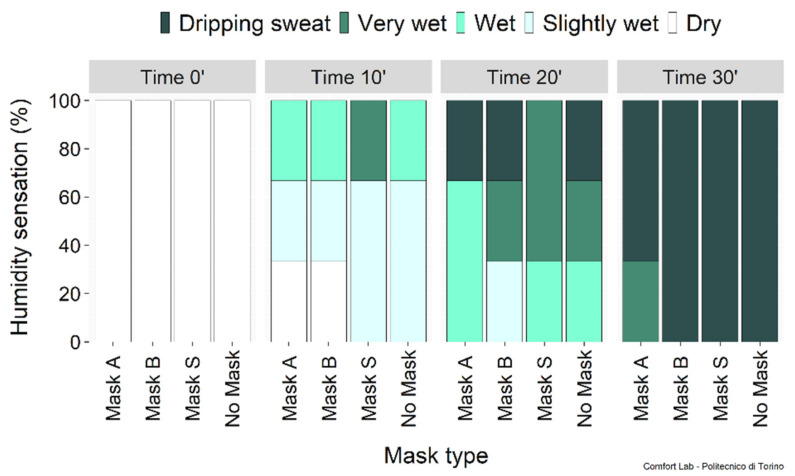
Perceived humidity sensation.

**Figure 13 ijerph-19-06491-f013:**
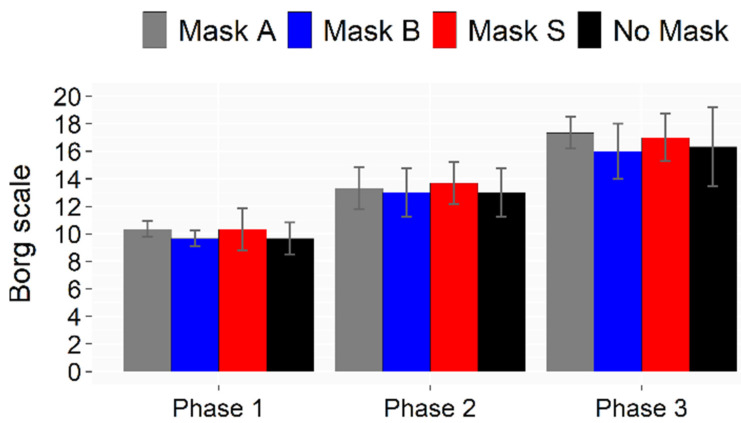
Rate of Perceived Exertion according to the Borg scale during the three phases of the test.

**Table 1 ijerph-19-06491-t001:** Biometric data of the involved testers. BMI is for body mass index; FTP is for functional threshold power.

Title 1	Tester 1	Tester 2	Tester 3
Age	22	22	22
Height (cm)	181	165	172
Weight (kg)	65	60	63
BMI ^1^	19.8	22.0	21.3
FTP ^2^ (W)	210	210	175

^1^ BMI = body mass index. ^2^ FTP = functional threshold power.

**Table 2 ijerph-19-06491-t002:** Characterization of the masks in terms of sound attenuation (∆SPL), sound absorption (α_0_), sound transmission loss (STL), breathing resistance, and filtration efficiency (*eCFC*). Standard deviations are given in the brackets.

	Mask A	Mask B	Mask S
∆SPL_0.4–5 kHz 3rd oct_ (dB)	2.9 (4.2)	1.0 (1.2)	0.4 (0.6)
∆SPL_1.6–5 kHz 3rd oct_ (dB)	6.1 (3.8)	1.8 (1.1)	0.8 (0.7)
α_0__, 1–5 kHz_ (-)	0.14 (0.08)	0.10 (0.07)	0.15 (0.07)
α_0__, 0.9–3 kHz_ (-)	0.18 (0.05)	0.13 (0.05)	0.18 (0.06)
STL_1–5 kHz_ (dB)	2.6 (0.6)	1.6 (0.2)	1.2 (0.1)
STL_1–3 kHz_ (dB)	2.8 (0.4)	1.6 (0.2)	1.3 (0.0)
Breathing Resistance (Pa)	353	123	66
*eCFC* (%)	98	93	87

**Table 3 ijerph-19-06491-t003:** Amount of moisture in the mask at the end of the test.

	Mask A	Mask B	Mask S
Average moisture content (g)	2.0 ± 0.5	1.7 ± 0.1	9.3 ± 0.4
Weight gain (%)	37.7	55.4	43.6

## Data Availability

The data presented in this study are available on request from the corresponding author.

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
