# Peer review of "Effects of Face Masks on Physiological Parameters and Voice Production during Cycling Activity"

_ijerph, 2022, doi:10.3390/ijerph19116491_

Round 1

Reviewer 1 Report

The authors present findings in respect with how the wearing of 3 masks affect the vocal parameters of cyclists, as well as other relevant parameters, such as thermal sensation (measured both in objective and subjective manners), heart rate, wettability, etc.

The manuscript is clearly written, but there are some minor issues that need to be addressed:

- The authors need to justify why were these three masks selected to be evaluated. It is more or less implied that they were selected to observe different designs which could be used for sport activities (since one is designed for this purpose, while the others for general use), but this should be explicit.

- The author should present the BPM data for heart rate (section 3.2.3), even if it is limited.

- The conclusions are quite surprising from an acoustics point of view. There are appears to be a difference in SPL when wearing a mask and not, but there aren't many differences of vocal parameters. SPL is literally measuring the intensity how air moves, which should directly affect how sound propagates. How can this be? Can the microphone distance be influenced by this? Or is the way that vocal parameters were measured make them somehow unrelated to the manner in which the SPL was measured? The authors should expand on this in Section 3.4.

Author Response

The authors present findings in respect with how the wearing of 3 masks affect the vocal parameters of cyclists, as well as other relevant parameters, such as thermal sensation (measured both in objective and subjective manners), heart rate, wettability, etc.

The manuscript is clearly written, but there are some minor issues that need to be addressed:

Thank you for the overall comment. We have addressed the comments in the sections below. All the changes that were made are highlighted in red color in the new version of the manuscript.

- The authors need to justify why were these three masks selected to be evaluated. It is more or less implied that they were selected to observe different designs which could be used for sport activities (since one is designed for this purpose, while the others for general use), but this should be explicit.

A specification was added from line 115 on.

- The author should present the BPM data for heart rate (section 3.2.3), even if it is limited.

A figure (now: Figure 10) representing the measured heart rate data was added in paragraph 3.3.3 and was appropriately introduced in the text.

- The conclusions are quite surprising from an acoustics point of view. There are appears to be a difference in SPL when wearing a mask and not, but there aren't many differences of vocal parameters. SPL is literally measuring the intensity how air moves, which should directly affect how sound propagates. How can this be? Can the microphone distance be influenced by this? Or is the way that vocal parameters were measured make them somehow unrelated to the manner in which the SPL was measured? The authors should expand on this in Section 3.4.

The method we used to measure voice parameters allows to consider the vocal effort exerted due to the vibration of the vocal folds as voice is measured in contact at the neck and not in air. The microphone distance is thus not relevant. A detailed description of the voice measurements procedure was given in paragraph 2.3 and particularly in the 2.3.1 section. We have added a further brief detailed paragraph in the results (Section  3.2) from line 411 on to answer to this comment.

Reviewer 2 Report

The paper contains the results of some effects of face masks. The test procedures are well described and presented. The main problem is a small number of testers. There were only three males of the same age. Two masks are suitable for team sports, whereas the third one is a slightly worse.

I cannot state that the work is original and novel, but at the end of the pandemic, it may be interesting to someone.

If it is decided to accept the manuscript, some points should be clarified:

  1. According to Table 2 the values of STL are ~2.7 dB for the mask A and ~1.5 dB for the masks B and S. The difference is only about 1 dB. Why do you state that the mask A is worse?
  2. What are the reasons to measure the absorption coefficient? I hope, you do not plan to use it for evaluation of sound reflection by a human head with a mask.
  3. The mean value Δ[email protected] for the mask A is much smaller the standard deviation. It means that the deviation of the measured parameters is too great. Is averaging carried out by both frequencies and testers? It would be useful to show spectra of Δ
  4. To characterize the voice transmission through the mask STI (speech transmission index) could be useful and applied along with other parameters.

Author Response

The paper contains the results of some effects of face masks. The test procedures are well described and presented. The main problem is a small number of testers. There were only three males of the same age. Two masks are suitable for team sports, whereas the third one is a slightly worse.

I cannot state that the work is original and novel, but at the end of the pandemic, it may be interesting to someone.

We thank the reviewer for the encouraging comment. Of course, the number of testers is the main drawback of the work, but since the work was performed during the pandemic, it was difficult to involve a large number of participants assuring the safety limitations introduced in the investigations with human subjects. However, we believe that this preliminary work might be of interest as useful reference regarding the multidisciplinary methodology and may be scalable at different contexts.  We have addressed the comments in the sections below, trying to make the revised manuscript improved. All the changes that were made are highlighted in red color in the new version of the manuscript.

If it is decided to accept the manuscript, some points should be clarified:

According to Table 2 the values of STL are ~2.7 dB for the mask A and ~1.5 dB for the masks B and S. The difference is only about 1 dB. Why do you state that the mask A is worse?

Mask A is worse since the STL is higher of about 1 dB, and the experimental standard deviations make this value significative different than the values of masks B and S. We do not claim more than this, e.g., that Mask A is the best mask ever. It is only a comparative result of one parameter among others. Furthermore, 1 dB is the JND of more acoustical parameters that are related to perception.

What are the reasons to measure the absorption coefficient? I hope, you do not plan to use it for evaluation of sound reflection by a human head with a mask.

The evaluation of the absorption coefficient was considered an important parameter to correlate to the outcomes of the study. In this framework, the main hypothesis was to consider the property of the material of mask as a design parameter that can be easily evaluated to predict the final acoustic performance in a similar way as the breathing resistance and filtration efficiency could help to predict the degree of protection performance.

The mean value Δ[email protected] for the mask A is much smaller the standard deviation. It means that the deviation of the measured parameters is too great. Is averaging carried out by both frequencies and testers? It would be useful to show spectra of Δ

The difference in SPL is evaluated with the Head and Torso Simulator in the anechoic room, thus no averaging has been made considering the tester. A brief clarification was added in the text from line 377 on and we agree with the reviewer on the usefulness of showing the spectra of Δ, so a figure (new Figure 5) was added.

To characterize the voice transmission through the mask STI (speech transmission index) could be useful and applied along with other parameters.

The Speech Transmission Index (STI) is a measure that is dependent on several factors and that cannot qualify the effective difference among masks. Among other aspects, STI is strongly dependent on the acoustics of the environment, as well as on the talker-to-listener distance, and thus the characterization of the face masks through this parameter would not give outputs useful in any case to the aim of the present study.

Reviewer 3 Report

Reviewer report for manuscript number ijerph-1700470

Thank you for giving me an opportunity to review this paper.

I only have some comments and questions as below.

Introduction

Investigating the impact of face masks on phonation and physical activities is necessary. Currently, COVID has come to a different stage where the use of face masks may not be mandatory in many settings e.g. outdoor sport. So it is not relevant to mention COVID as a lead-in basis for the paper.

This section should clarify more on why the impact of face masks was examined in this sport context. Why was voice outcome investigated in combination with physiological outcomes? More literature is needed to support the rationale of investigating physiological parameters whilst wearing a face mask. The aims of the study should be stated more clearly.

Materials and Methods

  1. The selection of face mask: Please clarify why the authors used two standard surgical masks given that these masks may have similar filtering characteristics. Would it be scientifically sound if a mask with higher filtering characteristics (e.g. N95, KN95) was used?
  2. It would be helpful if the authors gave a summary of filtering characteristics of the three mask types used in this study as provided by manufacturers.
  3. The number of participants was rather limited. This would affect interpretation of the findings.
  4. Please explain more clearly why a contact microphone was used to evaluate the effects of these face masks on voice production.
  5. Please clarify why SPL was estimated using the VHM via a calibration protocol rather than using a SPL meter. Does this protocol ensure accuracy of the SPL data?
  6. Please justify why the authors investigated the following outcome measures and how these would be related to the use of these face masks: breathing resistance and filtration efficiency.

Results

  1. What was the main focus of the paper (voice outcome or physiological outcome)? How were voice outcomes related to physiological outcomes in the context of this sport activity? Please present the most important/focused data first. The layout of the Result section made it difficult to understand what the main focus of the paper was.
  2. For a significant statistical test, please provide the test value (e.g. the result of the KW test), not only the p value (for example, line 431-432).
  3. Some findings would need more detailed description for better clarification rather than just presenting the information in tables and graphs.

Discussion

I would prefer reading a separate discussion section. In this paper the findings were not interpreted satisfactorily in terms of the relationship between wearing these mask types and the observed voice and physiological data. More detailed discussion would be needed in light of the literature.

Conclusion

As there were only three subjects, it would be hard to make strong claims on masks B and S in this sport activity.

Author Response

Reviewer report for manuscript number ijerph-1700470

Thank you for giving me an opportunity to review this paper. I only have some comments and questions as below.

We thank the reviewer for the comments and questions that allowed us to make changes for, we hope, the improvement of the quality, clarity and effectiveness of the manuscript. All the changes that were made are tracked in red color in the new version of the paper.

Introduction

Investigating the impact of face masks on phonation and physical activities is necessary. Currently, COVID has come to a different stage where the use of face masks may not be mandatory in many settings e.g. outdoor sport. So it is not relevant to mention COVID as a lead-in basis for the paper.

We have revised the abstract, the introduction and the conclusions according to the present comment. We have softened the link with the spreading of Covid-19, however we have kept a reference to it as it has been the origin of the need to start this experimental research.

This section should clarify more on why the impact of face masks was examined in this sport context. Why was voice outcome investigated in combination with physiological outcomes? More literature is needed to support the rationale of investigating physiological parameters whilst wearing a face mask. The aims of the study should be stated more clearly.

As suggested, we have included more studies in the introduction (in particular, from line 71 on) and revised the text to emphasize the rationale for investigating physiological parameters while wearing a face mask (from line 110 on).

Materials and Methods

The selection of face mask: Please clarify why the authors used two standard surgical masks given that these masks may have similar filtering characteristics. Would it be scientifically sound if a mask with higher filtering characteristics (e.g. N95, KN95) was used?

The Authors selected two surgical masks because, during the pandemic, the authorities indicated these devices as adequate to reduce the spreading of the virus. They were commonly used for everyday practices. Therefore, being readily available but less effective in removing particles than N95 respirators, the Authors decided to evaluate the effects of these mask typologies on voice production and physiological outcomes. Those results are compared with those obtained with the ad-hoc designed cloth mask for sport.

It would be helpful if the authors gave a summary of filtering characteristics of the three mask types used in this study as provided by manufacturers.

The information about fractional removal efficiency is not available for surgical masks because manufacturers rate them through their bacterial filtration efficiency. In contrast, those masks were measured with another approach in this study. Bacterial filtration efficiency and fractional filtration efficiency cannot be correlated directly. The sport masks are prototypes of community face coverings (not surgical masks or respirators), and the manufacturer provides the data reported in this study.

The number of participants was rather limited. This would affect interpretation of the findings.

We are aware of this limitation, and we have underlined that this is a pilot study in several points.

Please explain more clearly why a contact microphone was used to evaluate the effects of these face masks on voice production.

The reason for this is given at the beginning of paragraph 2.3.1., which reports “To avoid the effect of the environment on voice signal acquisitions and to allow for wearability of a monitoring device within the activity performance, as suggested in literature [31][34], a portable contact-sensor-based system was used to record voice production in this study.” We have added a clarification sentence, anyway, from line 195 on.

Please clarify why SPL was estimated using the VHM via a calibration protocol rather than using a SPL meter. Does this protocol ensure accuracy of the SPL data?

An air microphone is sensitive to the background noise, whereas a contact microphone is not. The protocol adopted in the present study, which is based on the use of a contact sensor to acquire the voice signal, ensures accuracy and traceability of the acquired data. The calibration procedure is briefly described in the manuscript (paragraph 2.3.1). However, detailed descriptions were avoided as we believe that they fall out of the scope of the present paper. Anyway, several previous studies are mentioned that can give more details on the procedure itself.

Please justify why the authors investigated the following outcome measures and how these would be related to the use of these face masks: breathing resistance and filtration efficiency.

We added a sentence to give answer to this comment from line 281 on.

Results

What was the main focus of the paper (voice outcome or physiological outcome)? How were voice outcomes related to physiological outcomes in the context of this sport activity? Please present the most important/focused data first. The layout of the Result section made it difficult to understand what the main focus of the paper was.

We thank the reviewer for this important comment. The main focus of the paper is related to the effect of face masks on voice parameters while performing intense physical activity. This aspect has been clarified more in the introduction; then, the results were adapted to present outcomes on voice first, and then outputs on physiological aspects. Some minor changes were also introduced in the text to underline the focus.

For a significant statistical test, please provide the test value (e.g. the result of the KW test), not only the p value (for example, line 431-432).

When possible, we have added the test coefficients (i.e., Z-values) in section 3.4, further than the p-values already reported.

Some findings would need more detailed description for better clarification rather than just presenting the information in tables and graphs.

As suggested, we have tried to add comments and references to discuss the obtained outcomes.

Discussion

I would prefer reading a separate discussion section. In this paper the findings were not interpreted satisfactorily in terms of the relationship between wearing these mask types and the observed voice and physiological data. More detailed discussion would be needed in light of the literature.

The possible comparisons with past studies are not extensive. Therefore, we believe that keeping the discussion section together with the results one is more effective. The conclusion section highlights the significant outcomes that bring to useful indications for everyday practice.

Conclusion

As there were only three subjects, it would be hard to make strong claims on masks B and S in this sport activity.

We have addressed this comment by softening some conclusions.

Round 2

Reviewer 2 Report

I agree with the authors – there are a lot of papers studying masks published last years. It would be nice to publish another one.

I would like only to pay attention at the recent paper in JASA https://doi.org/10.1121/10.0010384. It contains more detailed study of masks, and, probably, their results may be reflected in the reviewed paper.

Author Response

We thank the reviewer for mentioning this paper which has been publish few weeks ago. Reading it, some important aspects resulted to be correlated with the outcomes of the present study. Therefore, we have included the paper (now: reference n. 19).

Reviewer 3 Report

I thank the authors for their attempt to revise the paper. I have a few more comments to improve the quality of this paper as follows:

1. Please add the specific trade name/brand name of the surgical masks. 

2. Characteristics of the surgical masks were not provided as required. You mentioned "EN 14683", so how did these mask comply with this standard in terms of filtering characteristics? I am sure characteristics of surgical masks are available elsewhere.

3. Why you used two surgical masks was not explained satisfactorily. Why did you not use one surgical mask, or three masks? Why two?

4. Microphone. Although this contact microphone was not sensitive to environmental noise, was it affected by vibration or movement related to sport activities or during physical movement? If yes, did it affect the reliability of the recorded voice signals?

5. Recording. Did you have information about sampling rate? If yes, please include.

6. How equivalent/comparable were the SPL results obtained using this contact microphone compared to those recorded using a condenser 'air' microphone? A condenser microphone would record the voice signals as transmitted via the mouth/air conduction, so the signals would include both the source and vocal tract information. Would it be accurate to estimate the SPL information (which is affected by vocal tract filtering effects) using a contact microphone given that it did not capture the air-conducted signals? Did the data (Figure 7) really reflect the impact of the masks on the voice signals?

7. The first paragraph of Results can be put in Introduction. One would expect the first paragraph of Results to show how the results/findings would be presented.

Author Response

I thank the authors for their attempt to revise the paper. I have a few more comments to improve the quality of this paper as follows:

Thank you. Again, changes are tracked in red color in the revised version of the manuscript.

  1. Please add the specific trade name/brand name of the surgical masks.

We know the names of the masks’ resellers but there is no information about their manufacturers. Our intent was not to rate a degree of quality of the masks, selected among specific ones, but to establish a best practice based on a trade-off between filtration and acoustic performances.

  1. Characteristics of the surgical masks were not provided as required. You mentioned "EN 14683", so how did these mask comply with this standard in terms of filtering characteristics? I am sure characteristics of surgical masks are available elsewhere.

The masks we considered belong to a stock of face masks that our laboratories at Politecnico di Torino received at the beginning of the pandemic in early April 2020. The distributors of those masks did not provide any technical file or certified performance, still unavailable elsewhere. Our testing laboratory was required to assess their filtration performance to understand whether they could be considered truly as surgical masks. So far, the bacterial filtration efficiency required by EN 14683 and the test method implemented in our lab (based on UNI/PdR-90:2020) do not provide immediately comparable results.

  1. Why you used two surgical masks was not explained satisfactorily. Why did you not use one surgical mask, or three masks? Why two?

The reason why we decided to consider in this study two surgical masks, further than the ad-hoc textile one, is that our laboratories at Politecnico di Torino have received the formal request to evaluate a number of face masks at the very beginning of the pandemic in early April 2020. At that time, manufacturers converted their productions to face the health emergency needs, therefore Politecnico received successive stocks of face masks on which filtration tests had to be performed. Among the masks we received, we decided to select surgical masks that were easily and commonly distributed. Then, the criteria on which we selected the masks were: (i) an adequate filtration efficiency in agreement with EN 14683, (ii) a different breathing resistance performance, (iii) a different acoustic attenuation performance. Combining these three criteria, the two masks used in the study were the ones that better represented our needs and were thus used in comparison with the textile mask ad-hoc designed for sports at Politecnico. We added a clarification sentence on this in the methodology section (lines 139-143).

  1. Microphone. Although this contact microphone was not sensitive to environmental noise, was it affected by vibration or movement related to sport activities or during physical movement? If yes, did it affect the reliability of the recorded voice signals?

Previous studies have investigated the reliability of voice monitoring results based on workers performing their everyday activities, which also included body movements. The microphone has a not negligible sensitivity to body movements, but their effects are made negligible by means of a digital low-pass filter (implemented if the device firmware) that is able to separate the frequency components related to the voice activity from the ones related to the body movements. This specification was added in paragraph 2.3.1.

  1. Recording. Did you have information about sampling rate? If yes, please include.

With respect to the acquisition of the vocal signal, the sampling rate is set to 22050 Sa/s. With this sample frequency, the acquired samples were grouped into frames with a length of about 46.4 ms (1024 samples) as this value is of the same order of magnitude of the average inter-syllabic pause in the Italian language. We have added this information in the revised paper (lines 218-220).

  1. How equivalent/comparable were the SPL results obtained using this contact microphone compared to those recorded using a condenser 'air' microphone? A condenser microphone would record the voice signals as transmitted via the mouth/air conduction, so the signals would include both the source and vocal tract information. Would it be accurate to estimate the SPL information (which is affected by vocal tract filtering effects) using a contact microphone given that it did not capture the air-conducted signals? Did the data (Figure 7) really reflect the impact of the masks on the voice signals?

Extended literature is available on the techniques that can be used for voice monitoring. The use of contact-sensors to this aim is well established and is proved to be reliable in the estimation of voice parameters. They can well detect the impact of the masks on the voice signals: wearing a mask implies an increased exertion and fatigue that are reflected in the physical mechanisms adopted while speaking and thus in the vocal folds’ behavior. As far as the equivalency in SPLs in air and SPLs obtained with a contact-sensor-based methodology is concerned (investigated, e.g., by the authors in previous works like https://doi.org/10.1121/1.5042761 and https://doi.org/10.1121/1.5027816), it is even better to use the latter when a voice monitoring occurs in real conditions (e.g., in real environmental conditions like schools, offices, theaters; and in real behavioral conditions like while working, moving, etc). We have added specifications on these issues (lines 200-207), and generally we highlight that references are mainly related to the work by Titze and Svec, but also to that by some of the authors of the present paper.

  1. The first paragraph of Results can be put in Introduction. One would expect the first paragraph of Results to show how the results/findings would be presented.

We apologize for the inconvenience, but it was an unwanted copy-paste of the first paragraph already present in the introduction. We have removed it and we thank the reviewer for noticing this.